# Universal Representation of Generalized Convex Functions and their Gradients

**Moeen Nehzati** [1]

## Abstract

A wide range of optimization problems can often be written in terms of generalized convex functions (GCFs). When this structure is present, it can convert certain nested bilevel objectives into single-level problems amenable to standard first-order optimization methods. We provide a new differentiable layer with a convex parameter space and show (Theorems 4.3 and 4.10) that it and its gradient are universal approximators for GCFs and their gradients. We demonstrate how this parameterization can be leveraged in practice by (i) learning optimal transport maps with general cost functions and (ii) learning optimal auctions of multiple goods. In both these cases, we show how our layer can be used to convert the existing bilevel or min-max formulations into single-level problems that can be solved efficiently with first-order methods.

## 1. Introduction

This paper targets a common need in machine learning: principled, differentiable parameterizations that encode structure beyond generic universal approximators. We study generalized convex functions (GCFs) and their gradients, and provide a practical parameterization with universal approximation guarantees and a convex parameter space.

Generic shallow and deep neural networks (SNNs and DNNs) can approximate a wide variety of functions, but they are not the best tool for all tasks. Their flexibility can come at the cost of losing important structural properties and requiring more data. In many applications we instead exploit symmetries or shape constraints. For instance, translation-invariant image classifiers are modeled with convolutional neural networks (CNNs), which are universal approximators for translation-invariant functions (Yarotsky, 2022).

Convex functions and their gradients are two particularly useful structured classes that have recently received increased attention from a parameterization perspective. In practice, however, many objects of interest are not convex but share key properties with convex functions. Generalized convexity relaxes convexity to capture precisely this broader structure (see Singer (1997) for a survey).

Generalized convexity extends classical convexity by replacing the bilinear pairing in convex conjugation with an application-dependent kernel/surplus $\Phi(x, y)$. Section 3 gives the formal definition. Two motivating examples in this paper are optimal transport and mechanism design: in optimal transport we work with the surplus $\Phi(x, y) = -c(x, y)$ (the negative of the cost of moving mass from $x$ to $y$) and the Kantorovich dual potentials can be chosen GCFs; in quasilinear mechanism design, the buyer's indirect utility is a GCF where $\Phi(x, y)$ is the value a buyer of type $x$ gets from outcome $y$. Section 5 reviews these connections and their role in bilevel formulations.

Despite advances in parameterizing convex functions and their gradients, relatively little work has addressed GCFs. As a result, when learning GCFs, existing methods often ignore their structural properties, especially in bilevel and min–max settings such as optimal auctions with multiple goods and optimal transport with general costs. These problems are typically much harder to solve numerically than single-level optimization, and the lack of structure-aware parameterizations has limited the scope of problems that can be tackled with theoretical guarantees.

The goal of this paper is to close this gap by developing universal approximators for GCFs and their gradients and demonstrating how this theoretical machinery can be used in practice. We show that our parameterization can recover and extend classical convex-analytic constructions while remaining amenable to gradient-based training in modern ML workflows.

**Contributions.** We summarize our main contributions:

- A differentiable parameterization of GCFs with a convex parameter space, enabling first-order optimization.
- Universal approximation results for both GCFs and their gradients under mild regularity conditions on the cost/surplus kernel.

[1]Department of Economics, New York University, New York, NY, USA. Correspondence to: Moeen Nehzati <moeen.nehzati@nyu.edu>.

*Proceedings of the 43rd International Conference on Machine Learning*, Seoul, South Korea. PMLR 306, 2026. Copyright 2026 by the author(s).

- A neural-network interpretation that connects finitely $Y$-convex parameterizations to shallow architectures with max aggregation, suggesting deeper analogues.
- An open-source implementation with experiments on multi-item auction design and optimal transport that instantiate the theory.

The remainder of the paper is organized as follows. Section 2 reviews related work on parameterizing convex functions and their gradients. Section 3 introduces convexity and generalized convexity. Section 4 presents the main theoretical results on the parameterization and universal approximation of GCFs and their gradients. Section 5 reviews how GCFs arise in optimal transport and mechanism design and Section 6 presents empirical results on these applications using the proposed parameterization. We conclude in Section 7 with some closing remarks.

**Conflict of Interest Disclosure** The author declares no financial conflicts of interest.

## 2. Related Work

The effectiveness of neural networks is partly due to their Universal Approximation Property (UAP): any sufficiently regular function can be approximated by a large enough neural network, whether shallow or deep, a fact studied extensively in, e.g., Hornik et al. (1989); Pinkus (1999); Liang & Srikant (2016); Lu et al. (2021).

Closer to our context is the literature on parameterizing and approximating convex functions. Perhaps the most natural scheme is the max-affine parameterization: any convex function can be represented as the supremum of possibly infinitely many affine functions (its subgradients). Choosing the maximum of finitely many affine functions underlies max-affine regression, as explored in Balázs et al. (2015). Calafiore et al. (2019) and Kim & Kim (2022) show how the maximum can be replaced with the Log-Sum-Exp (LSE) function to yield smooth approximations. Other works, such as Warin (2023) and Amos et al. (2017), propose more sophisticated multi-layered parameterizations, while Magnani & Boyd (2009) study piecewise linear convex functions.

Another line of research concerns the approximation and parameterization of gradients. In contrast to the one-dimensional case, not every vector field $f : \mathbb{R}^n \to \mathbb{R}^n$ is the gradient of some scalar-valued function $g : \mathbb{R}^n \to \mathbb{R}$. When $g$ is smooth, a necessary condition for $f = \nabla g$ is the Jacobian $J_f$ being symmetric, since it equals the Hessian $H_g$, which can be hard to impose. A naive idea is to parametrize gradients by differentiating parametrizations of scalar functions (for example, by using derivatives of neural networks to approximate derivatives of functions), but this approach can fail (Saremi, 2019). Even if $f_n \to f$, it does not necessarily follow that $\nabla f_n \to \nabla f$; for example:

$$f_n(x) = \frac{1}{n}\sin(nx) \to 0 \text{ yet } f'_n(x) = \cos(nx) \not\to 0. \quad (1)$$

These problems make UAP results for gradients less common and much harder to obtain. As discussed in Section 4, this difficulty disappears when the functions (and their limits) are convex; Chaudhari et al. (2024) uses this fact to construct universal approximators for gradients of convex functions. Richter-Powell et al. (2021) and Lorraine & Hossain (2024) pursue a different approach by parameterizing the second derivative (the symmetric positive definite Hessian) and integrating it via neural ordinary differential equations Chen et al. (2018a).

On the practical side, such parameterizations are routinely used to learn convex objects end-to-end: (i) convex potentials whose gradients yield transport maps or Wasserstein barycenters (Makkuva et al., 2020; Fan et al., 2020); (ii) convex value/energy functions embedded in model-based control and optimization loops (Chen et al., 2018b); (iii) convex potentials that parameterize generative flows grounded in optimal transport (Huang et al., 2020); and (iv) convex functionals over probability measures optimized directly (Alvarez-Melis et al., 2021). In these setups, "finding something convex" is the goal by design, making ICNNs and related architectures a natural fit. Complementarily, differentiable convex optimization layers embed convex programs within networks, enabling end-to-end training with convex structure (Amos & Kolter, 2017; Agrawal et al., 2019), and differentiable MPC implements convex optimization-based control policies in a learnable manner (Amos et al., 2018). More broadly, deep declarative networks provide a unifying view of embedding optimization problems as differentiable layers (Gould et al., 2021).

In contrast to convexity, computational aspects of generalized convexity remain less explored. For surveys of the mathematical theory, see van De Vel (1993); Pallaschke & Rolewicz (2013); Singer (1997); Rubinov (2013). GCFs are ubiquitous in many areas of applied mathematics, particularly in optimal transport, matching, and game theory. After introducing GCFs in Section 3, we showcase some of their applications in Section 5. While there has been substantial work on parameterizations of convex functions and their gradients, far less has been done for GCFs. We provide analogous results for parameterization and approximation of GCFs, and establish universal approximation guarantees for generalized convex functions and their gradients (Theorems 4.3 and 4.10).

On the mechanism design or pricing side, some works jointly learn a mechanism together with a model of buyer behavior, leading to bilevel training procedures (Rahme

et al., 2020; Nedelec et al., 2019; Shen et al., 2018). Other approaches rely on multi-agent reinforcement learning to learn mechanisms and agent policies simultaneously, which also induces a bilevel structure (Zheng et al., 2022; Koster et al., 2022).

On the OT side, the computational difficulties arise from enforcing the marginal constraints: in the primal formulation, the transport plan must have prescribed marginals, while in the dual formulation the potentials must satisfy the Kantorovich inequality. To avoid handling these constraints explicitly, several works embed them into unconstrained min-max objectives. On the primal side, Rout et al. (2021) learn a transport map through an adversarial consistency objective that implicitly enforces the pushforward constraint. On the dual side, Makkuva et al. (2020); Korotin et al. (2022b;a) introduce min-max formulations in which the Kantorovich feasibility constraint is encoded directly in the objective through neural parameterizations of the dual potentials.

## 3. Background

### 3.1. Convexity

Let $V$ be a Euclidean space and $V^*$ its dual. Take $g : V^* \to \overline{\mathbb{R}}$ to be an extended-real-valued function. Since we work in Euclidean spaces, we identify $V^{**}$ with $V$. Under this identification, the Legendre transform (convex conjugate) of $g$ can be viewed as a function on $V$ defined by

$$g^* : V \to \overline{\mathbb{R}} \qquad g^*(x) = \sup_{y \in V^*} \{\langle x \mid y \rangle - g(y)\}$$

The Legendre transform is central to convex analysis: (under standard regularity conditions) a function $f : V \to \overline{\mathbb{R}}$ is convex if and only if it coincides with the Legendre transform of another function: $f = g^*$.

### 3.2. Generalized Convexity

Since the Legendre transform is defined via a supremum over affine functions, a natural generalization replaces the bilinear pairing with a more general bivariate function. We replace $V$ and $V^*$ with sets $X$ and $Y$ (typically subsets of Euclidean spaces). We then define the 'kernel' $\Phi : X \times Y \to \mathbb{R}$ to take place of the pairing.

Everything that follows depends on the choice of kernel $\Phi$ but we suppress it in the notation for brevity.

We also allow the supremum to be taken over a subset $\tilde{Y} \subseteq Y$ (throughout, a tilde indicates an arbitrary subset). When $g$ is defined on $\tilde{Y}$, we denote its $\tilde{Y}$-transform by a superscript $\tilde{Y}$:

$$g^{\tilde{Y}} : X \to \overline{\mathbb{R}} \qquad g^{\tilde{Y}}(x) = \sup_{y \in \tilde{Y}} \{\Phi(x, y) - g(y)\}$$

When $\Phi(x, y) = \langle x \mid y \rangle$ and $\tilde{Y} = Y$, these definitions recover the classical Legendre transform (in the classical case one takes $Y = X^*$). Unlike standard convexity theory, we allow for functions defined on a subset of $X$. We say a function $f$ is $\tilde{Y}$-convex if it is the restriction of a $\tilde{Y}$-transform to the domain of $f$:

$$f = (g^{\tilde{Y}})_{|\mathrm{dom}(f)}$$

Finally, denote the set of $\tilde{Y}$-convex functions defined on $\tilde{X}$ by $\mathcal{C}^{\tilde{Y}}(\tilde{X})$. Define $\tilde{X}$-transform, $\tilde{X}$-convexity, and $\mathcal{C}^{\tilde{X}}(\tilde{Y})$ analogously.

Appendix A.1 proves some known properties of generalized convexity analogues to those of standard convexity. Those properties are used in the proofs of theorems in later sections.

## 4. Method: Parameterizing Generalized Convex Functions

Due to the symmetry between $X$ and $Y$, we focus on parameterizing $\mathcal{C}^Y(X)$, the space of $Y$-convex functions defined over $X$. We assume that the surplus $\Phi$ is locally Lipschitz. From now on, we also assume $X$ and $Y$ are compact; therefore $\Phi$ is (globally) Lipschitz on $X \times Y$. In particular, $x \mapsto \Phi(x, y)$ is Lipschitz with a constant uniform in $y$. Hence any $Y$-convex function is Lipschitz, since it is the supremum of a family of Lipschitz functions sharing the same Lipschitz constant.

We define a function to be finitely $Y$-convex if it is the $\tilde{Y}$-transform of some function where $\tilde{Y} \subseteq Y$ is finite (recall: a tilde indicates an arbitrary subset). Denote the space of all finitely $Y$-convex functions defined over $X$ by $\mathcal{FC}^Y(X)$:

$$\mathcal{FC}^Y(X) = \bigcup_{\tilde{Y} \subseteq Y \wedge |\tilde{Y}| < \infty} \mathcal{C}^{\tilde{Y}}(X)$$

Parameterizing $\mathcal{FC}^Y(X)$ is straightforward. Fix a finite $\tilde{Y}$. For this fixed support set, the space $\mathcal{C}^{\tilde{Y}}(X)$ is parameterized by the intercept values $p \in \mathbb{R}^{\tilde{Y}}$. In practice, we also treat the support locations as parameters, writing $\tilde{Y} = \{y^1, \ldots, y^n\}$ with $(y^1, \ldots, y^n) \in Y^n$ and optimizing jointly over $(y^1, \ldots, y^n, p_1, \ldots, p_n)$. We additionally assume $Y \subseteq \mathbb{R}^d$ is convex so that $Y^n \times \mathbb{R}^n$ is a convex parameter space. This convexity assumption is only used for the convex-parameter-space claim; it is not needed for the approximation results below.

Our first result shows that finitely $Y$-convex functions can uniformly approximate $Y$-convex functions.

**Proposition 4.1** (Uniform approximation of $Y$-convex functions). *Given any $\epsilon > 0$, there is a finite $\tilde{Y} \subseteq Y$ such that for any $Y$-convex function $f \in \mathcal{C}^Y(X)$, there exists*

$g \in \mathcal{C}^{\tilde{Y}}(X)$ *such that*

$$|f - g|_\infty < \epsilon$$

*Proof.* Deferred to Appendix (A.2.1).

It would be convenient if finitely $Y$-convex functions were also $Y$-convex. This is not a given; for example, rationals are not irrationals yet they can be arbitrarily close to them. The following proposition takes care of that.

**Proposition 4.2.** *Finitely $Y$-convex functions are also $Y$-convex.*

$$\mathcal{FC}^Y(X) \subseteq \mathcal{C}^Y(X)$$

*Proof.* Deferred to Appendix (A.2.1).

Let $\overline{S}$ denote the topological closure of $S$. Combining Propositions 4.1 and 4.2 we obtain:

**Theorem 4.3** (Density of finitely $Y$-convex functions). *The finitely $Y$-convex functions are dense in the space of $Y$-convex functions:*

$$\overline{\mathcal{FC}^Y(X)} = \mathcal{C}^Y(X)$$

*Hence, our parameterization of $\mathcal{FC}^Y(X)$ is a universal approximator for $\mathcal{C}^Y(X)$.*

*Proof.* Deferred to Appendix (A.2.1).

This may not be enough, as sometimes we need to approximate the gradients of $Y$-convex functions. For example, in Section 5, we will see that allocations depend on the gradient of the GCF indirect utility function.

The example of equation 1 in Section 2 demonstrated that uniform convergence of functions does not imply convergence of their gradients. The reason is that controlling values of functions is not enough to control their local slopes. As shown in the literature (see Chaudhari et al. (2024)), this problem goes away when the functions and their limit are convex. For convex functions we have the following property:

$$\exists p \forall z: \ f(z) \geq f(x) + \langle z - x \mid p \rangle \qquad (2)$$

We call such $p$ subgradients of $f$ at $x$ and they coincide with gradients where $f$ is differentiable. Using this, for any $v$, we obtain:

$$\frac{f(x) - f(x - tv)}{t} \leq \langle v \mid p \rangle \leq \frac{f(x + tv) - f(x)}{t}. \qquad (3)$$

Under uniform convergence $f_n \to f$, these finite-difference bounds converge. Since the inner product is point-separating ($p \mapsto \langle \cdot \mid p \rangle$ is injective), controlling $\langle v \mid p \rangle$ for all $v$ controls $p$ itself, which in turn controls the gradients where $f$ is differentiable.

Moving from convexity to generalized convexity, we have the following generalization of equation 2:

$$\exists p \forall z: \ f(z) \geq f(x) + \Phi(z, p) - \Phi(x, p) \qquad (4)$$

Such $p$ are called $\Phi$-subgradients of $f$ at $x$. There are two non-trivial obstacles to applying the same logic here: (1) $\Phi$ may fail to be point-separating in its second argument, and (2) in spite of their name, $\Phi$-subgradients do not coincide with gradients of $f$ and may not even be in a one-to-one correspondence with them.

Though we did not mention it then, the counterexample equation 1 can be realized within generalized convexity. Take $X = Y = [-2\pi, 2\pi]$ and the continuous kernel

$$\Phi(x, y) = \begin{cases} y \sin(x/y), & y \neq 0, \\ 0, & y = 0. \end{cases}$$

For each $n$, let $\tilde{Y}_n = \{1/n\}$ and define $g_n : \tilde{Y}_n \to \mathbb{R}$ by $g_n(1/n) = 0$. Then $f_n(x) = g_n^{\tilde{Y}_n}(x) = \Phi(x, \frac{1}{n}) = \frac{1}{n}\sin(nx)$. Since each $f_n$ is finitely $Y$-convex, it is also $Y$-convex by Proposition 4.2. Similarly, letting $\tilde{Y} = \{0\}$ realizes $f \equiv 0$ as a $Y$-convex function.

So uniform convergence of GCFs does not imply convergence of their gradients. Intuitively, at least in this counterexample, the obstruction is unbounded curvature: $f_n''(x) = -n\sin(nx)$ has magnitude $n \to \infty$. This matters as curvature determines how quickly the gradients change. Hence we cannot have something analogous to equation 3.

To recover a kernel-agnostic substitute for this convex mechanism, we identify an abstract condition ensuring a uniform lower curvature bound for all branches $x \mapsto \Phi(x, y)$, which then propagates through suprema and restores gradient stability. We formalize this via semiconvexity, defined next.

**Definition 4.4** (Semiconvexity). A function $f : X \to \overline{\mathbb{R}}$ is semiconvex if there exists a constant $K \geq 0$ such that $f + \frac{K}{2}\|x\|_2^2$ is convex. A family of functions is equi-semiconvex if they are all semiconvex with the same constant $K$.

*Remark* 4.5. In parts of the optimization literature, this definition is also referred to as $K$-weak convexity. We use the term "semiconvexity" because it is more common in the generalized-convexity and optimal-transport literature.

*Remark* 4.6. If $f$ is twice differentiable on $X$, then $K$-semiconvexity is equivalent to $\nabla^2 f(x) \succeq -KI$ for all $x$ (i.e., every Hessian eigenvalue is $\geq -K$).

See Cannarsa & Sinestrari (2004) for a thorough introduction.

This is a much weaker condition than convexity since sufficiently smooth functions are semiconvex on a compact domain. Intuitively, semiconvexity only requires the absence of downward kinks, since any finite negative curvature can

be compensated by adding a sufficiently large quadratic term.

Going back to our counterexample equation 1, notice that $f_n$s are $n$-semiconvex while $f$ is 0-semiconvex, so semiconvexity alone is not sufficient. As it turns out, semiconvexity would be enough if $f_n$s and $f$ shared the same semiconvexity constant, which we call equi-semiconvexity.

**Proposition 4.7** (Stability of gradients under semiconvex convergence). *If $f_n \to f$ uniformly and all $f_n$ and $f$ are equi-semiconvex, then $\nabla f_n \to \nabla f$ uniformly where the gradients exist.*

*Proof.* Deferred to Appendix (A.2.1). ∎

This result allows us to pass from uniform approximation of functions to uniform approximation of their gradients within an equi-semiconvex family, and it is the key bridge between function-level and gradient-level universal approximation in our setting.

Hence, the natural question is: Are $Y$-convex functions equi-semiconvex?

**Proposition 4.8** (Preservation of semiconvexity under $\tilde{Y}$-transform). *If the functions $\Phi(\cdot, y)$ are equi-semiconvex, then every $\tilde{Y}$-convex function is semiconvex with the same constant (and thus $\mathcal{C}^{\tilde{Y}}(X)$ is an equi-semiconvex family).*

*Proof.* Deferred to Appendix (A.2.1). ∎

*Remark* 4.9. A sufficient condition is that $\Phi$ is twice continuously differentiable since by compactness of $X \times Y$, we can lower bound the smallest eigenvalue of $\nabla_x^2 \Phi(x, y)$ uniformly.

In many applications, boundedness/compactness of $X$ and $Y$ is natural (or can be imposed without affecting the modeling goal, e.g., by normalization or truncation). Moreover, in applications where one aims to recover decision rules from a GCF via twist/envelope-type formulas (as in Section 5), one typically assumes enough regularity of the kernel (often $C^2$ on the relevant compact domain) so that these expressions are well-defined. Under these common compactness and smoothness assumptions, the equi-semiconvexity condition required for our gradient-approximation results is satisfied by the preceding remark.

**Theorem 4.10** (Universal approximation for gradients). *If the sections $x \mapsto \Phi(x, y)$ are equi-semiconvex, then $\nabla \mathcal{FC}^Y(X) = \{\nabla f : f \in \mathcal{FC}^Y(X)\}$ is dense in $\nabla \mathcal{C}^Y(X) = \{\nabla f : f \in \mathcal{C}^Y(X)\}$:*

$$\overline{\nabla \mathcal{FC}^Y(X)} = \nabla \mathcal{C}^Y(X)$$

*In other words, $\nabla \mathcal{FC}^Y(X)$ are universal approximators for $\nabla \mathcal{C}^Y(X)$.*

*Proof.* Deferred to Appendix (A.2.1). ∎

Since finitely $Y$-convex functions are defined as a finite maximum, they are not smooth. In certain applications, we may prefer to work with smoothed versions. In the standard convex setting, some recent works replace the maximum with the log-sum-exp function $\mathrm{LSE}^\tau$:

$$\mathrm{LSE}^\tau(x_1, \ldots, x_n) = \frac{1}{\tau} \ln \left( \sum_{i=1}^n e^{\tau x_i} \right).$$

To define a smoothed version of the $\tilde{Y}$-transform, we replace the maximum with $\mathrm{LSE}^\tau$ and call it the $\tilde{Y}^\tau$-transform. For a function $g : \tilde{Y} \to \mathbb{R}$, define

$$g^{\tilde{Y}^\tau}(x) = \frac{1}{\tau} \ln \left( \sum_{y \in \tilde{Y}} \exp \left( \tau(\Phi(x, y) - g(y)) \right) \right).$$

Similarly, define $\mathcal{FC}^{Y^\tau}(X)$ and $\nabla \mathcal{FC}^{Y^\tau}(X)$ by replacing the $\tilde{Y}$-transform with the $\tilde{Y}^\tau$-transform.

**Theorem 4.11** (Smooth approximation). *$\bigcup_{\tau \in \mathbb{N}} \mathcal{FC}^{Y^\tau}(X)$ uniformly approximates $\mathcal{C}^Y(X)$. If the kernel sections $x \mapsto \Phi(x, y)$ are equi-semiconvex, then $\bigcup_{\tau \in \mathbb{N}} \nabla \mathcal{FC}^{Y^\tau}(X)$ pointwise approximates $\nabla \mathcal{C}^Y(X)$ where gradients exist.*

*Proof.* Deferred to Appendix (A.2.1). ∎

*Remark* 4.12. Theorem 4.11 justifies replacing the hard max with log-sum-exp to obtain smooth models while retaining some approximation guarantees (and, under equi-semiconvexity, gradient approximation where gradients exist).

### 4.1. Maxout Analogy

This subsection provides intuition on finitely $Y$-convex models by relating their structure to maxout-type architectures.

Our finitely $Y$-convex parameterization takes the form

$$x \mapsto \max_{i=1,\ldots,n} \{\Phi(x, y^i) - p_i\},$$

that is, it aggregates a collection of branch functions by $\max$. In the inner-product case $\Phi(x, y) = \langle x, y \rangle$, the branches are affine in $x$, and the model reduces to a maxout layer. Concretely, a maxout unit computes the maximum of finitely many affine functions, e.g.

$$x \mapsto \max_{j=1,\ldots,k} \{\langle w_j \mid x \rangle - b_j\}.$$

For general $\Phi$, the branches $\Phi(\cdot, y^i) - p_i$ are kernel sections and need not be affine. Figure 2 in the appendix illustrates this.

This viewpoint suggests exploring deeper compositions of finitely $Y$-convex modules as a complementary direction to increasing the number of support points $n$, but we do not pursue this here.

# 5. Applications of Generalized Convexity

We will show how certain optimal transport and auction design problems can be framed as finding the right generalized convex function.

## 5.1. Optimal Transport

An optimal transport problem concerns relating a distribution of mass $\mu \in \Delta(X)$ on one space to a distribution of mass $\eta \in \Delta(Y)$ on another space in a cost-minimizing way.

Since minimizing a cost is equivalent to maximizing its negation, we shall work with the surplus kernel $\Phi(x, y) = -c(x, y)$ and frame the problem as maximization.

$$\sup_{\pi \in \Pi(\mu, \eta)} \mathbb{E}_\pi[\Phi(x, y)] \tag{5}$$

$$\inf_{f: X \to \mathbb{R}, \, g: Y \to \mathbb{R}} \mathbb{E}_\mu[f(x)] + \mathbb{E}_\eta[g(y)] \tag{6}$$

$$\text{s.t.} \quad \forall (x, y): \quad f(x) + g(y) \geq \Phi(x, y)$$

Equation 5 states the Kantorovich problem: finding a transportation plan $\pi \in \Pi(\mu, \eta)$, where a coupling $\Pi(\mu, \eta)$ is the set of all joint distributions on $X \times Y$ with marginals $\mu$ and $\eta$. Since this is linear, it also admits the dual in Equation 6, where $f, g$ are called the Kantorovich potentials.

For a feasible $(f, g)$ pair, $(g^Y, g)$ is also feasible (see Theorem A.1) and does not increase the dual objective. Doing it one more time yields $(g^Y, (g^Y)^X)$, hence it is WLOG to write the dual as an optimization over GCFs:

$$\inf_{f \in \mathcal{C}^Y(X)} \mathbb{E}_\mu[f(x)] + \mathbb{E}_\eta[f^X(y)] \tag{7}$$

Another key result from the literature is that

$$\pi(x, y) > 0 \implies \nabla f(x) = \nabla_x \Phi(x, y)$$

When $\nabla_x \Phi(x, \cdot)$ is a diffeomorphism (which in particular satisfies the twist condition[1]), we can invert it to get the optimal transportation map $T: X \to Y$:

$$T(x) = (\nabla_x \Phi(x, \cdot))^{-1}(\nabla f(x)) \tag{8}$$

This is a generalization of (Brenier, 1991), which states that for $c(x, y) = \|x - y\|_2^2$ the optimal transport map is the gradient of a convex function.

## 5.2. Mechanism Design

Let there be outcomes $Y$, each priced according to a pricing function $t: Y \to \mathbb{R}$. A buyer has a type $x \in X$ unknown to the seller affecting how much they value each outcome. Buyer's utility $u(x, y) = \Phi(x, y) - t(y)$ is the value $\Phi(x, y)$

---

[1] Twist means that for each fixed $x$, the map $y \mapsto \nabla_x \Phi(x, y)$ is injective, so $(\nabla_x \Phi(x, \cdot))^{-1}$ is well-defined on its range.

they get from the outcome $y$, given their type $x$, minus their payment $t(y)$. Under the compactness assumption in Section 4, we can write the buyer's choice problem as:

$$Q(t, x) = \arg\max_{y \in Y} u(x, y) = \arg\max_{y \in Y} \Phi(x, y) - t(y)$$

The seller, on the other hand, receives a revenue $t(y)$ minus some production cost $P(y)$ with $P: Y \to \mathbb{R}$ where $y$ should be the outcome chosen by the buyer. Hence the objective of the seller is:

$$\sup_{t: Y \to \mathbb{R}} \mathbb{E}_X(t(Q(t, x)) - P(Q(t, x)))$$

$$\text{s.t.} \quad Q(t, x) = \arg\max_{y \in Y} u(x, y)$$

In other words, the seller needs to find a price function that gives them high profit given that the buyer is making the optimal choice $Q(t, x)$. From an ML perspective, this is exactly an adversarial training or min–max setup: the seller chooses $t$ while anticipating the buyer's best response $Q(t, x)$. As mentioned in Section 1, this bilevel formulation is what many applied papers use. However, we will show that using GCFs, we can reduce this to a much simpler single-level optimization problem. To do that, we need to first define the indirect utility function, that is the utility the buyer receives from making their best choice.

$$v(x) = \max_{y \in Y} u(x, y) = \max_{y \in Y} \Phi(x, y) - t(y)$$

It is easy to see that $v$ is a $Y$-convex function, i.e., $v \in \mathcal{C}^Y(X)$. We will show how we can write the bilevel problem as a single-level problem in terms of $v$. For that, we need the revelation principle from mechanism design (Myerson, 1981). Simply put, it states that for any mechanisms with prices $t$ and choices $Q$, there is a simpler equivalent mechanism that removes the optimization problem on the buyer's end.

Given prices $t$ and choices $Q$, we can ask the buyer directly for their types and simulate their optimal choice $Q(t, x)$ for them. This simplifies the buyer's side as the buyer has no incentive to report anything but their true type. Hence, we can convert the buyer's strategic optimization into a constraint on the seller's side. These mechanisms are called direct revelation incentive compatible (DRIC) mechanisms. Direct revelation means the buyer is only asked for their type and incentive compatible (IC) means their optimal choice is to reveal its true value. The distinction may seem superficial at first but it will allow us to rewrite the problem.

As discussed, it is WLOG to work with DRIC mechanisms so we focus on them. A DRIC mechanism consists of two objects: a payment function $p: X \to \mathbb{R}$ and an allocation function $a: X \to Y$ deciding what outcome to assign to each reported type. We will show that given a GCF indirect

utility $v$, (IC) pins down both the payment $p$ and allocation $a$.

In the finitely $Y$-convex model, we can recover an allocation rule by selecting an active support point: pick $i^*(x) \in \arg\max_i \{\Phi(x, y^i) - p_i\}$ and set $a(x) := y^{i^*(x)}$.

When $v$ is differentiable, we can apply the envelope theorem (Rochet, 1987) [2] to the indirect utility $v$ to get:

$$\nabla v(x) = \nabla_x \Phi(x, a(x))$$

Again when $\nabla_x \Phi(x, \cdot)$ is invertible (the twist condition mentioned before), we have $a(x) = (\nabla_x \Phi(x, \cdot))^{-1}(\nabla v(x))$, so the allocation is pinned down by the gradient of the indirect utility $v$. To simplify the notation, define $W(v, x) = (\nabla_x \Phi(x, \cdot))^{-1}(\nabla v(x))$ so $a(x) = W(v, x)$. We can also write the payment in terms of the indirect utility from its definition:

$$p(x) = \Phi(x, a(x)) - v(x) = \Phi(x, W(v, x)) - v(x)$$

Thus, both the payment and the allocation are determined by the indirect utility function $v$, and we can write our bilevel problem as that of choosing the right GCF, the indirect utility function $v$:

$$\max_{0 \le v \in \mathcal{C}^Y(X)} \mathbb{E}_X [\Phi(x, W(v, x)) - v(x) - P(W(v, x))] \tag{9}$$

The constraint $0 \le v$ is there to make sure the buyer would want to participate in the auction. More concretely, it means $\forall x : 0 \le v(x)$, otherwise the buyer with type $x$ would not want to participate in the auction. See (Ekeland, 2010) for a more detailed discussion.

# 6. Experiments

These experiments should be read as structured nonlinear optimization benchmarks rather than standard supervised learning tasks. In optimal transport, one can apply OT to many data modalities (including images), but this requires choosing a cost/kernel $\Phi$ that encodes the application semantics; in most such datasets there is no canonical $\Phi$, and different choices lead to different problems. Since our goal is to study learning potentials/maps under an explicit, user-specified $\Phi$ (including non-quadratic/general costs), we avoid conflating kernel selection with the contribution here. In mechanism design, the situation is even starker: there is no standard suite of public "real-world" benchmark datasets with agreed valuation models/kernels and ground-truth optima. We therefore focus on controlled benchmarks where $\Phi$ is explicit and where either partial baselines exist (Setting A) or meaningful sanity/qualitative checks are available

(OT, Setting B). Importantly, we include non-inner-product kernels in both domains (general-cost OT potentials and a nonlinear auction surplus in Setting B).

The code for the experiments are available as part of the `gconvex` package that implements finitely convex functions and optimizes them using PyTorch.

## 6.1. Implementation Details

In both applications, the learned object is a GCF: in optimal transport, it is the Kantorovich dual potential $f$; in mechanism design, it is the indirect utility $v$ (Section 5). In both cases, we parameterize the GCF as a finitely $Y$-convex model (i.e., a $\tilde{Y}$-transform with $\tilde{Y} = \{y^1, \dots, y^n\} \subseteq Y$):

$$x \mapsto \max_{i=1,\dots,n} \{\Phi(x, y^i) - p_i\},$$

with trainable support points $\{y^i\}_{i=1}^n \subseteq Y$ and intercepts $\{p_i\}_{i=1}^n$. There is no separate choice of approximation layer beyond this parameterization: once the kernel $\Phi$ is fixed, the main approximation knob is the number $n$ of support points.

Training objectives are those in Section 5, with expectations approximated by finite-sample averages, and optimized with first-order methods (Adam/AdamW). When constraints are present, we enforce them through penalty terms. To ease the optimization, we replace the hardmax in the objective with a softmax relaxation and ramp up the temperature over the course of training to approach the hardmax. When evaluating, we use the hard-max (and we observed that using a softmax at evaluation yields nearly identical results once the temperature is sufficiently high).

We recover downstream objects from the learned GCF using the characterizations in Section 5. In OT, our smooth costs satisfy the twist condition, hence we recover the Monge map via the twist-based formula equation 8. In mechanism design, we recover the allocation rule by an argmax over the learned support points $\{y^i\}$; this does not rely on twist and is essential in Setting B, whose hinge surplus does not satisfy twist. We then set payments via $p(x) = \Phi(x, a(x)) - v(x)$ (Section 5.2).

## 6.2. Optimal Transport

Looking back at Equation 7, with our approach solving the Kantorovich problem is contingent on $f^X$ being easy to compute. For example, in the 1-Wasserstein case (metric cost), Kantorovich–Rubinstein duality avoids explicit conjugation by reducing the dual to a single 1-Lipschitz potential (one may take $g = -f$). Alternatively, if the marginals are product measures and the cost is additively separable, tensorization converts the $n$-dimensional conjugation to $n$ one-dimensional ones, almost trivial to solve (see Villani et al. (2008) for more information on both). Here, we focus

---

[2]Intuitively, when the maximizer $y^*(x)$ is unique and in the interior (so the first-order condition in $y$ applies), differentiating through the max gives $\nabla F(x) = \nabla_x f(x, y^*(x))$.

on the latter case. We use a mixture of measures and consider two different costs: the quadratic cost $\|x - y\|_2^2$ and its negative $-\|x - y\|_2^2$. Though these look similar, they lead to very different results as the former prefers to transport by minimal displacement while the latter prefers to transport by maximal displacement. In particular, computing optimal transport with non-convex costs such as $-\|x - y\|_2^2$ is beyond the abilities of most solvers. On compact domains these costs are smooth, hence $\nabla_x^2 \Phi(x, y)$ admits a uniform lower bound, so the kernel sections are equi-semiconvex.

Figure 1 visualizes the results. It's easy to see that the marginals are well matched. Additionally, with the quadratic cost, the transport map is not moving the mass too far while with the negative quadratic cost, the transport map is moving the mass as far as possible. This is in line with what the costs are incentivizing.

## 6.3. Auction Design

For auction design experiments, we will use the formulation in Equation 9 to express the seller's profit in terms of the GCF indirect utility $v$. To pin down a problem we need to specify 5 things: (1) outcome space $Y$, (2) type space $X$, (3) surplus kernel $\Phi$, (4) the production cost $P$ and lastly (5) the distribution of types. The problem with this and similar exercises is that there are very few known baselines. Settings where the optimal auction is known are limited and there are not many numerical benchmarks to compare against for more than two goods. Hence we will consider two settings: (A) a simpler setting where we have some idea about what the optimal auction should look like and (B) a more complex setting where theory can't buy us much but there are sanity checks the results should satisfy. Setting A uses the classical inner-product surplus and therefore reduces to the standard convex-analytic setting (rather than a genuinely generalized convex one); we include it because it is the regime with the strongest available baselines. Note that even in this classical case, the training problem over finitely many support points and intercepts is nonconvex. Setting B (and the nonstandard OT costs above) illustrates the non-inner-product setting.

Before moving on, we note that computational mechanism design becomes challenging quickly as the number of goods grows. Even for the classical additive/inner-product surplus (Setting A), recent learned-auction benchmarks typically consider on the order of 10–15 goods (some allow for a handful of buyers rather than one) (Dütting et al., 2019; Curry et al., 2022b; Ivanov et al., 2022; Duan et al., 2023; Curry et al., 2022a). In contrast, in Setting A we report results for a single buyer with up to 250 goods. Additionally, we study a more complex Setting B with a non-inner-product kernel and report results up to 20 goods.

*Table 1.* Mean profit per good in Setting (A) (small $n$). SJa revenue is reported when available.

| Profit per Good | $n = 1$ | $n = 2$ | $n = 4$ | $n = 6$ | $n = 12$ |
|---|---|---|---|---|---|
| Separate Posted Pricing (analytical) | 0.250 | 0.250 | 0.250 | 0.250 | 0.250 |
| SJA | 0.250 | 0.274 | 0.305 | 0.324 | 0.361 |
| Learned Mechanism | 0.249 | 0.274 | 0.303 | 0.321 | 0.353 |

### 6.3.1. SETTING A

We set $X = Y = [0, 1]^n$, $\Phi(x, y) = \langle x, y \rangle$, $P(y) = 0$. We can interpret this as follows. There is a single buyer and $n$ goods for sale. The types $x \in X$ denote how much the buyer values each good. The outcomes $y \in Y$ denote the probability of receiving each good. The kernel $\Phi(x, y)$ gives the expected value a buyer of type $x$ receives from outcome $y$. There are no production costs so the seller is not incurring any loss by transferring the good. Even though this setting may seem simple, it's hard to approach theoretically. Almost all the theory is concerned with the types being uniformly distributed in $X = [0, 1]^n$ so we take the distribution of types to be uniform as well. Here the kernel sections are affine in each variable, hence 0-semiconvex and equi-semiconvex.

We know that as $n$ grows, the seller can do better by bundling some of the goods together instead of selling them separately. However, we do not know how exactly this bundling should be done and what the optimal profit is. The Straight-Jacket auction (SJa) is known to be optimal for $n \leq 6$ (Giannakopoulos & Koutsoupias, 2014) and conjectured to be optimal for $6 < n$. Even computing the revenue of this auction is complicated. Joswig et al. (2022) provides its exact revenue for up to $n \leq 12$, so we compare against that where available. Tables 1–2 summarize our main results: Table 1 reports SJa for small $n$, while Table 2 focuses on larger $n$ where no exact SJa benchmark is available and we report only analytical baselines and our learned mechanism. We are able to closely match SJa's revenue where available, providing a quantitative check. In addition, we observe two sanity checks. The seller can always attain profit per item of 0.25 by selling each good separately at price 0.5. As the number of items grows, the seller should be able to do better via bundling of goods. The profit per item can never exceed 0.5 as in that case the buyer would be better off not participating in the auction. Our findings are consistent with these expectations as the profit per good increases with the number of goods while staying within the bounds of $[0.25, 0.5]$.

Figure 3 from appendix visualizes the learned allocations for the two-item case. Similar to SJa, the learned mechanism is neither pure bundling nor separate selling. Instead, it's a combination of both selling goods separately and together.

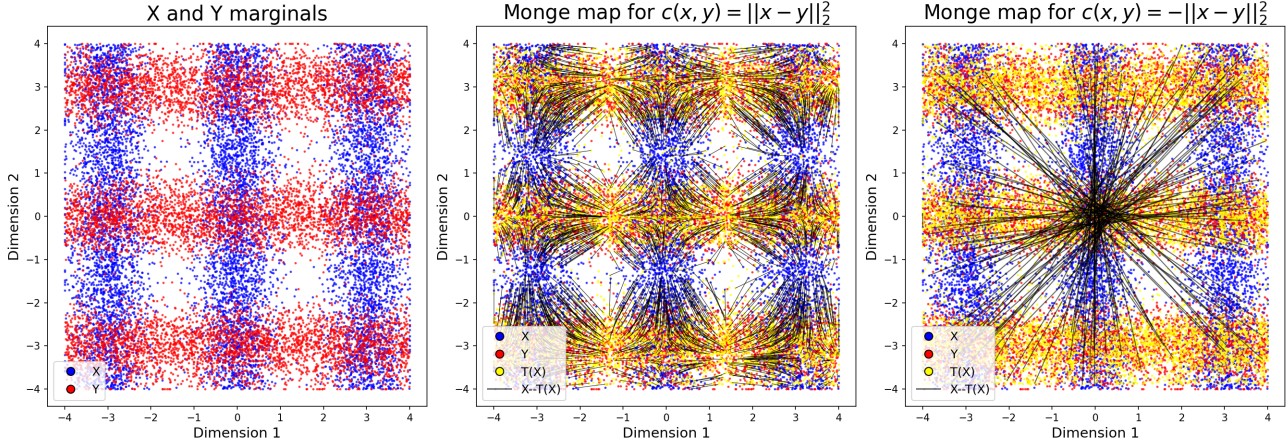

*Figure 1.* Visualization of the transport maps. The left plot shows the samples from the marginals. The middle and right plot show the underlying marginals and $T(X)$ where $T$ is the optimal learned transport map for $\|x - y\|_2^2$ and $-\|x - y\|_2^2$ respectively. The black lines connect $X$ and $T(X)$. As expected, the lines in the middle plot are short while lines in the right plot are as long as possible.

*Table 2.* Mean profit per good in Setting (A) (large $n$).

| Profit per Good | $n = 50$ | $n = 100$ | $n = 250$ |
| --- | --- | --- | --- |
| Separate Posted Pricing (analytical) | 0.250 | 0.250 | 0.250 |
| Learned Mechanism | 0.412 | 0.423 | 0.450 |

*Table 3.* Mean profit per good in Setting (B).

| Profit per Good | $n = 1$ | $n = 5$ | $n = 10$ | $n = 20$ |
| --- | --- | --- | --- | --- |
| Learned Mechanism | 1.022 | 1.202 | 1.250 | 1.253 |

### 6.3.2. SETTING B

Let $X = Y = [0, T]^n$ and $\Phi(x, y) = \sum_{i=1}^n \max(y_i - x_i, 0)$. We assume types $x$ are i.i.d. log-normal with parameters $(0, 0.25)$ truncated to $[0, T]^n$. We take $T$ sufficiently large so that for the finite sample sizes used in our experiments, the truncation does not bind in practice. We set production cost $P(y) = 0.1\|y\|_2^2$. One interpretation is that $Y$ is the quantity of each good produced and sold while $X$ is the need of the buyer for each good. The surplus kernel $\Phi(x, y)$ captures how well the produced quantity $y$ meets the need $x$ of the buyer, giving the buyer no value if the need is not met and more if it is exceeded. Here the kernel sections are convex in each variable, hence 0-semiconvex and equi-semiconvex.

Table 3 summarizes our results. The surplus is separable across goods (so it does not model complementarities), but it is nonlinear and falls outside the inner-product case. Since there are no known optimal mechanisms or strong numerical baselines here, we focus on sanity checks: as the number of goods increases, the profit per good increases (bundling benefits), and the profit per good remains non-negative and bounded above by the expected value of a single good under the given distribution.

## 7. Conclusion

We developed a framework for parameterizing generalized convex functions and their gradients with a convex parameter space. Finitely $Y$-convex functions form a dense subset of all $Y$-convex functions, yielding universal approximators for both generalized convex functions and their gradients under mild regularity conditions. These parameterizations admit a neural-network interpretation via shallow architectures with max aggregation, suggesting deeper analogues.

On the applied side, our methods are implemented in the `gconvex` package and used to learn optimal transport maps with general costs and revenue-maximizing auctions with general valuation kernels, with results consistent with theory. This provides a foundation for further work in mathematical economics, optimal transport, and bilevel ML, including designing deeper finitely $Y$-convex architectures, understanding when local optimization finds global optima, developing quantitative approximation rates in the number of support points beyond density guarantees, and analyzing convergence behavior of the resulting nonconvex training dynamics.

## Acknowledgements

I thank Alfred Galichon and Pegah Alipoormolabashi for helpful discussions and comments.

## Impact Statement

This paper is primarily theoretical and studies generalized convex parameterizations together with controlled simulated applications to optimal transport and mechanism design. In particular, the mechanism design settings considered here are stylized mathematical abstractions of economic interactions. Any mechanism that is "optimal" in this framework is optimal only relative to the specified primitives and objective, such as the surplus kernel, type distribution, production cost, and seller revenue criterion; this does not imply fairness, transparency, or broader social desirability. The paper does not study deployment questions or ethical constraints, and the experiments are limited to controlled synthetic settings rather than real-world decision systems.

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

# A. Appendix

## A.1. Useful Properties of Generalized Convexity

**Theorem (Basic properties).** If $f, g : \tilde{X} \to \mathbb{R}$, then for all $(x, y) \in \tilde{X} \times \tilde{Y}$:

$$\Phi(x, y) \leq f^{\tilde{X}}(y) + f(x) \qquad f^{\tilde{Y}\tilde{X}}(x) \leq f(x) \qquad f \leq g \implies f^{\tilde{X}} \geq g^{\tilde{X}}$$

*Proof.* By definition of the $\tilde{X}$-transform,

$$f^{\tilde{X}}(y) = \sup_{x' \in \tilde{X}} \Phi(x', y) - f(x')$$
$$\geq \Phi(x, y) - f(x),$$

which yields the first inequality.

For the second property,

$$f^{\tilde{Y}\tilde{X}}(x) = \sup_{y \in \tilde{Y}} \Phi(x, y) - f^{\tilde{X}}(y)$$
$$\leq \sup_{y \in \tilde{Y}} f(x) = f(x),$$

where the inequality follows from the first property.

For the third property, if $f \leq g$ then

$$g^{\tilde{X}}(y) = \sup_{x \in \tilde{X}} \Phi(x, y) - g(x)$$
$$\leq \sup_{x \in \tilde{X}} \Phi(x, y) - f(x) = f^{\tilde{X}}(y).$$

$\square$

A useful analogue to the invariance under biconjugation property of convex functions is the following:

**Theorem (Generalized biconjugation).** Let $f : \tilde{X} \to \mathbb{R}$. Then $f$ is $\tilde{Y}$-convex if and only if

$$f = (f^{\tilde{Y}\tilde{X}})_{|\tilde{X}}$$

*Proof.* If $f$ equals its generalized biconjugate on $\tilde{X}$ then it is, by definition, the transform of $f^{\tilde{X}}$ and hence $\tilde{Y}$-convex. Conversely, suppose $f$ is $\tilde{Y}$-convex so there is $g$ with $f = (g^{\tilde{Y}})_{|\tilde{X}}$. Applying the $\tilde{X}$-transform and then the $\tilde{Y}$-transform yields

$$f^{\tilde{X}} = (g^{\tilde{Y}})^{\tilde{X}} = g^{\tilde{X}\tilde{Y}} \leq g,$$

where the last inequality is implied by the Basic properties theorem above. Restricting back to $\tilde{X}$ and taking transforms gives

$$(f^{\tilde{Y}\tilde{X}})_{|\tilde{X}} \geq (g^{\tilde{Y}})_{|\tilde{X}} = f,$$

which together with the general inequality $f^{\tilde{Y}\tilde{X}} \leq f$ proves equality. $\square$

*Remark* A.1. Here the notation $f^{\tilde{Y}\tilde{X}}$ means that we first take the $\tilde{Y}$-transform and then the $\tilde{X}$-transform of $f$. Thus the identity $f = (f^{\tilde{Y}\tilde{X}})_{|\tilde{X}}$ is the natural analogue of $f = (f^{**})_{|X}$ in the classical convex-conjugate setting, and the order of transforms matches the usual biconjugation convention.

As a simple corollary, distinct $\tilde{Y}$-convex functions have distinct $\tilde{X}$-transforms.

## A.2. Proofs

### A.2.1. METHOD (SECTION 4)

**Proof of Proposition (Uniform approximation of $Y$-convex functions).** Since $\Phi$ has a Lipschitz constant $\lambda$, for any $y_1, y_2 \in Y$,

$$|y_1 - y_2| < \frac{\epsilon}{2\lambda} \implies |\Phi(x, y_1) - \Phi(x, y_2)| < \frac{\epsilon}{2}.$$

Since $f$ is $Y$-convex,

$$f(x) = f^{YX}(x) = \sup_{y \in Y}\{\Phi(x, y) - f^X(y)\},$$

where $f^X$ is $X$-convex and also has Lipschitz constant $\lambda$. Thus,

$$|y_1 - y_2| < \frac{\epsilon}{2\lambda} \implies |f^X(y_1) - f^X(y_2)| < \frac{\epsilon}{2}.$$

Under the compactness assumption in Section 4, $X \times Y$ is compact and metric, hence totally bounded. So we can cover $Y$ with finitely many balls of radius $\frac{\epsilon}{2\lambda}$; let the centers be $\tilde{Y} = \{y_1, \ldots, y_k\}$. Since $\tilde{Y} \subseteq Y$, we have $f^{\tilde{Y}X} \leq f^{YX} = f$. To show the reverse inequality, observe that any $y \in Y$ is within distance $\frac{\epsilon}{2\lambda}$ of some $y_i \in \tilde{Y}$, so

$$\Phi(x, y) - f^X(y) \leq \Phi(x, y_i) - f^X(y_i) + \epsilon.$$

Therefore,

$$\begin{aligned} f^{YX}(x) &= \sup_{y \in Y}\{\Phi(x, y) - f^X(y)\} \\ &\leq \sup_{y_i \in \tilde{Y}}\{\Phi(x, y_i) - f^X(y_i) + \epsilon\} \\ &= f^{\tilde{Y}X}(x) + \epsilon. \end{aligned}$$

$\square$

**Proof of Proposition (Finitely $Y$-convex functions are $Y$-convex).** Let $f \in \mathcal{C}^{\tilde{Y}}(X)$ for some finite $\tilde{Y} \subseteq Y$. By definition there is a function $g_1$ on $\tilde{Y}$ such that

$$f = (g_1^{\tilde{Y}})_{|X}.$$

Define a function $g_2$ on $Y$ by

$$g_2(y) = \begin{cases} g_1(y), & y \in \tilde{Y}, \\ +\infty, & y \in Y \setminus \tilde{Y}. \end{cases}$$

Then $g_2^Y$ coincides with $g_1^{\tilde{Y}}$ on $X$, so $f = (g_2^Y)_{|X}$ and hence $f \in \mathcal{C}^Y(X)$. Since $f$ was an arbitrary finitely $Y$-convex function, this shows $\mathcal{FC}^Y(X) \subseteq \mathcal{C}^Y(X)$. $\square$

**Proof of Proposition (Stability of gradients under semiconvex convergence).** We can rely on the results concerning convex functions. Define $h_n(x) = f_n(x) + \frac{K}{2}\|x\|_2^2$ and $h(x) = f(x) + \frac{K}{2}\|x\|_2^2$. Then $h_n$ and $h$ are convex, so $\nabla h_n \to \nabla h$ uniformly where defined. Since $\nabla f_n = \nabla h_n - \frac{K}{2}\nabla\|x\|_2^2$ and $\nabla f = \nabla h - \frac{K}{2}\nabla\|x\|_2^2$, we obtain $\nabla f_n \to \nabla f$ uniformly where the gradients exist. $\square$

**Proof of Proposition (Semiconvexity is preserved).** By equi-semiconvexity of the family $\{\Phi(\cdot, y)\}_{y \in Y}$, there exists $K \geq 0$ such that for every $y \in \tilde{Y}$ the function $x \mapsto \Phi(x, y) + \frac{K}{2}\|x\|_2^2$ is convex. Let $f \in \mathcal{C}^{\tilde{Y}}(X)$, so $f(x) = \sup_{y \in \tilde{Y}}\{\Phi(x, y) - g(y)\}$ for some $g$ on $\tilde{Y}$. Then

$$f(x) + \frac{K}{2}\|x\|_2^2 = \sup_{y \in \tilde{Y}}\left\{\Phi(x, y) + \frac{K}{2}\|x\|_2^2 - g(y)\right\},$$

which is a supremum of convex functions and hence convex. Therefore $f$ is semiconvex with constant $K$. $\square$

**Proof of Theorem (Density of finitely $Y$-convex functions).** By Proposition 4.1, for any $\epsilon > 0$ and any $f \in \mathcal{C}^Y(X)$ there is a finite $\tilde{Y}$ and $g \in \mathcal{C}^{\tilde{Y}}(X)$ with $\|f - g\|_\infty < \epsilon$. Proposition 4.2 shows $\mathcal{C}^{\tilde{Y}}(X) \subseteq \mathcal{C}^Y(X)$. Hence $\overline{\mathcal{FC}^Y(X)} = \mathcal{C}^Y(X)$. $\square$

**Proof of Theorem (Universal approximation for gradients).** Let $(g_n)_n \subset \mathcal{FC}^Y(X)$ uniformly approximate $f \in \mathcal{C}^Y(X)$ (Theorem above). Assume the family $\{\Phi(\cdot, y)\}_{y \in Y}$ is equi-semiconvex with constant $K$. By Proposition 4.8, each $g_n$ and $f$ are semiconvex with the same constant $K$. By Proposition 4.7, $\nabla g_n \to \nabla f$ uniformly where gradients exist. Thus $\overline{\nabla \mathcal{FC}^Y(X)} = \nabla \mathcal{C}^Y(X)$. $\square$

**Proof of Theorem (Smooth approximation).** Recall that for any real numbers $a_1, \ldots, a_m$ and $\tau > 0$,

$$\max_i a_i \leq \mathrm{LSE}^\tau(a_1, \ldots, a_m) \leq \max_i a_i + \frac{\log m}{\tau}.$$

Fix finite $\tilde{Y}$ and define $h(x) = \max_{y \in \tilde{Y}}\{\Phi(x, y) - g(y)\}$ and its smoothed version $h_\tau(x) = \mathrm{LSE}^\tau(\{\Phi(x, y) - g(y)\}_{y \in \tilde{Y}})$. Then $\|h_\tau - h\|_\infty \leq \frac{\log |\tilde{Y}|}{\tau}$. Combining with the uniform approximation of $\mathcal{C}^Y(X)$ by finitely $Y$-convex functions yields that $\bigcup_\tau \mathcal{FC}^{Y^\tau}(X)$ uniformly approximates $\mathcal{C}^Y(X)$. Moreover, when the family $\{\Phi(\cdot, y)\}_{y \in Y}$ is equi-semiconvex, each $h_\tau$ is smooth and semiconvex; as $\tau \to \infty$, $h_\tau \to h$ uniformly and $\nabla h_\tau \to \nabla h$ pointwise wherever the maximizer is unique (a.e. under mild conditions), giving pointwise density of gradients. $\square$

## A.3. Graphs

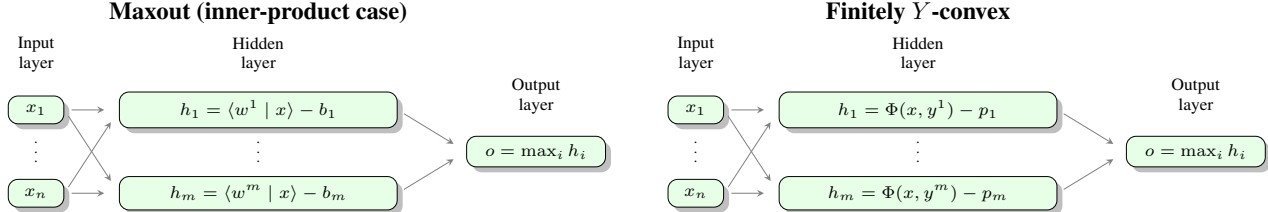

*Figure 2.* Maxout analogy for finitely $Y$-convex models. The left diagram depicts a maxout-type unit: affine branches $x \mapsto \langle w^i \mid x \rangle - b_i$ aggregated by $\max$. The right diagram depicts a finitely $Y$-convex model: kernel branches $x \mapsto \Phi(x, y^i) - p_i$ aggregated by $\max$. In the inner-product case $\Phi(x, y) = \langle x, y \rangle$, the right-hand model reduces to the left-hand max-affine form.

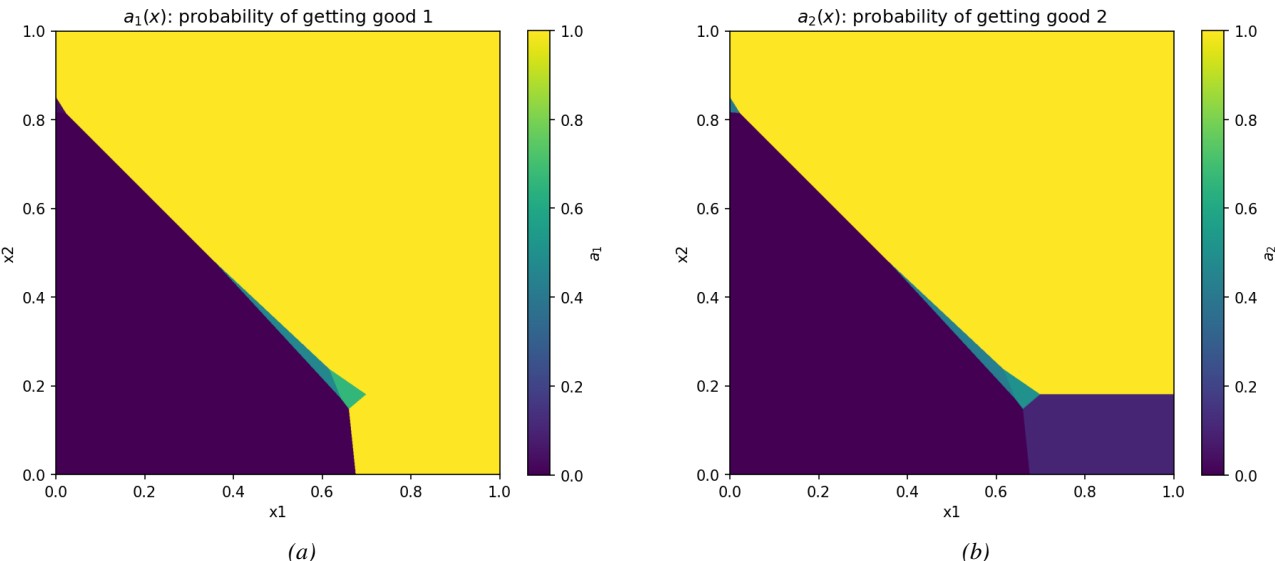

*(a)*                                         *(b)*

*Figure 3.* Auction learned for the two-item case for Setting (A). The allocation is neither pure bundling nor separate selling. Similar to SJa, it prices combinations of items and exhibits bunching.

