# OpenReview forum: "Universal Representation of Generalized Convex Functions and their Gradients"
_ICML.cc/2026/Conference — ICML 2026 regular_

### Official Review · Reviewer_5sfL · 2026-03-11

**Soundness:** 2
**Presentation:** 2
**Significance:** 3
**Originality:** 3
**Overall Recommendation:** 4
**Confidence:** 4

**Summary:**

This paper proposes a generalized convexity condition to accommodate applications beyond convexity. Under semiconvexity, this paper proves the universal approximation for both function values and gradients using a differentiable layers with convex parameter spaces. Moreover, the paper provides how to leverage these results by proving that optimal control with general cost functions and auction learning problem of multiple goods problems have this structure. Therefore, both of them can be solved by single level formulation with first order approaches. Numerical experiments validate the effectiveness of the proposed method.

**Compliance With Llm Reviewing Policy:**

Affirmed.

**Final Justification:**

The major concern is around the presentation in this paper. After the author's rebuttal, most of my previous concerns have been solved. I agreed that the theoretical contribution might be of interest to the optimal control community so I increased my score.

**Key Questions For Authors:**

See weekness. I think the major limitations lie in the clarity of the writing (especially the algorithm is not clear) and the insufficiency of both the theoretical and experimental results.

What is the formal definition of $a(y)$ and $T(y)$? It seems the authors only provide their calculations in terms of $\Phi$, but I'm confused about its original mathematical definition.

In experiment setting A, as you choose $\Phi$ as the inner product, will that make your problem convex?

The semiconvexity condition seems to be similar with weak convexity condition in optimization, e.g. see [R1].

**Limitations:**

Yes

**Strengths And Weaknesses:**

Strengths:

1. First of all, the proposed generalized convexity condition is novel and its verification on two applications of optimal control and auction learning problems are intriguing. The method appears to rely only on mild semiconvexity assumptions, which are relatively easy to satisfy in many settings, such as weakly convex functions [R1]. Overall, I find the core idea of the paper insightful, and the proposed condition innovative.

2. The numerical results on optimal control and auction learning effectively demonstrate the performance of the proposed algorithm.

[R1] Davis, Damek, and Dmitriy Drusvyatskiy. "Stochastic model-based minimization of weakly convex functions." SIAM Journal on Optimization 29.1 (2019): 207-239.

Weakness:

1. As a counterpart to the first strength, although the assumptions appear to be weak, the theoretical guarantees in the paper also seem somewhat limited. In particular, the results do not provide finite-time convergence guarantees and appear to omit certain error sources. For example, in practice, the optimal solution $a(y)$ is usually unavailable and must be estimated from the buyer’s side, which introduces a natural source of error. However, the theorem does not seem to capture the robustness of the method to such estimation error.

2. Related to the first concern, some parts of the paper writing are not entirely clear to me. For instance, what exactly is the algorithm proposed in this paper? My understanding is that the method may rely on differentiable layers to approximate the true problem, but I'm not sure how to choose those approximation layers. It would be much clearer if the paper explicitly described what is done at each step and included pseudocode for the algorithm.

3. The connection to neural networks in Section 5 is also unclear to me. While the setting seems to share some structural similarities with neural networks, the activation functions, linear layers, and other components appear quite different. I am therefore wondering whether the framework in the paper actually covers neural networks as a special case, and if so, what the corresponding $\Phi(\cdot)$ would be.

4. The final concern is the experimental scale. It seems that all the experiments are conducted on relatively small-scale problems with low-dimensional parameters, but please correct me if I'm wrong. I'm wondering how your algorithm performs in larger-scale experiments like neural network training.

---

> ### Author Rebuttal · Authors · 2026-03-24
>
> Thank you for the careful reading and for the helpful feedback. Below are the noted concerns.
>
> ## 1. Algorithmic pipeline is unclear
>
> I agree. The current paper does not state the experimental pipeline explicitly enough.
>
> In both applications, the learned object is a generalized convex potential $f$: in optimal transport, $f$ is the Kantorovich dual potential; in mechanism design, $f$ is the indirect utility (denoted by $v$ in Section~4.2). In both cases, we parameterize $f$ as a finitely $\Phi$-convex model
> $$
> f(z;\mathbf y,p)=\max_{i=1,\dots,n}\{\Phi(z,y^i)-p_i\}.
> $$
> with trainable support points $\mathbf y=(y^1,\dots,y^n)$ and intercepts $p=(p_1,\dots,p_n)$. There is no separate choice of approximation layer beyond this parameterization: once the kernel $\Phi$ is fixed, the main approximation knob is the number $n$ of support points.
>
> For optimization, I do not work with the hard max directly, but use a softmax relaxation; when constraints are present, I enforce them through penalty terms.
>
> Sections~4.1 and~4.2 write the application objectives as expectations in terms of $f$(in OT this includes the induced $\Phi$-transform $f^X$). In the experiments, these expectations are approximated by finite-sample averages and optimized with first-order methods. The task-specific outputs are then recovered from the learned potential.
>
> I will add a short implementation subsection and pseudocode to make this explicit.
>
> ## 2. Definitions of $a(y)$ and $T(x)$
>
> $T(x)$ denotes the transport map in OT, and $a(y)$ denotes the allocation rule in mechanism design. Both are induced by a feasible solution to the respective optimization problem. They are not primitive problem data and not independent optimization variables. I express them in terms of $\Phi$ because we are interested in the $T$ and $a$ induced by the optimal solution, and those objects are characterized in terms of $\Phi$.
>
> ## 3. Approximation guarantees versus convergence guarantees
>
> I agree that the paper does not provide finite-time convergence guarantees for the training algorithm, nor robustness guarantees for practical numerical or statistical error sources. Those are outside the current scope. The theoretical claims are approximation-theoretic: universal approximation for generalized convex functions, and gradient approximation when the kernel sections are equi-semiconvex.
>
> I would like to point out, however, that in our formulation $a(y)$ is not separately estimated from data; under the twist condition, it is recovered explicitly from the learned indirect utility $v$ via the twist/envelope formulas. I will revise the presentation to make this scope explicit.
>
> ## 4. Does Setting A reduce to the classical convex case?
>
> Yes. The reviewer is correct that when $\Phi(x,y)=\langle x,y\rangle$, the represented class reduces to the classical convex case. However, this does not make the parameter optimization problem convex: the training problem over finitely many support points and intercepts remains nonconvex. I included this setting because it is the regime with the strongest known baselines, so it is the most informative place to validate the method quantitatively.
>
> I agree that Setting A alone does not showcase the generalized aspect of the framework. That aspect is instead demonstrated by the non-inner-product experiments, including the nonlinear kernel in Setting B. I will revise the experimental discussion to make this distinction explicit.
>
> ## 5. Experimental scale
>
> Although larger-scale experiments would be valuable, the most relevant comparison here is to prior auction learning works, which are typically much smaller. In particular, recent learned-auction benchmarks cited usually study roughly $10$--$15$ goods, whereas in Setting A I report results for up to $250$ goods for a single buyer. Even in the more complicated Setting B, I report results for a non-inner-product kernel with up to $20$ goods.
>
>
> ## 6. Neural-network comparison
>
> I agree that the current neural-network discussion is too loose. The intended comparison is structural, not a claim that standard shallow networks with summation are a special case.
>
> More precisely, for fixed support points and intercepts, the model computes
> $$
> \max_i \Phi(x,y_i)-r_i,
> $$
> or a smoothed version. In the inner-product case $\Phi(x,y)=\langle x,y\rangle$, this becomes a max-affine / maxout-type shallow architecture. For general $\Phi$, the branch functions are application-specific and need not be affine. I will revise Section~5 accordingly.
>
> ## 7. Semiconvexity versus weak convexity
>
> In Hilbert spaces, which include the finite-dimensional Euclidean setting used in the paper, these notions coincide: a function is $K$-semiconvex if and only if it is $K$-weakly convex. I use the term "semiconvexity" because it is standard in the generalized-convexity / optimal-transport literature. I will add a short remark clarifying the relation to weak convexity.

---

> > ### Author Rebuttal · Reviewer_5sfL · 2026-04-03
> >
> > I thank the authors for the detailed response. Most of my concerns have been solved so I decided to increase my score to 4.

---

### Official Review · Reviewer_Gq1V · 2026-03-12

**Soundness:** 2
**Presentation:** 2
**Significance:** 3
**Originality:** 3
**Overall Recommendation:** 4
**Confidence:** 2

**Summary:**

The paper proposes a novel parametrization of generalized convex functions (also Y-convex functions) and their gradients by introducing the finitely Y-convex functions. It establishes that finitely Y-convex functions form a dense subset of Y-convex functions and provide uniform approximation guarantees. Under an additional equi-semiconvexity assumption, the paper further proves uniform approximation of gradients by the gradients of finitely Y-convex functions. The proposed parametrization is conceptually compared to fully connected layers in neural networks. The paper frames optimal transport and auction design problems as instances of learning generalized convex functions, and presents empirical results on both tasks that support the theoretical approximation claims.

**Compliance With Llm Reviewing Policy:**

Affirmed.

**Final Justification:**

The rebuttal convinced me to slightly increase my score.

**Key Questions For Authors:**

1. The paper introduces finitely Y-convex functions as a theoretical parametrization. How is the parametrization used in the experiments?

2. The uniform approximation of gradients relies on an equi-semiconvexity assumption. Are there meaningful examples or applications (including those in the experiments) where this assumption is violated, and if so, how does the method behave?

**Limitations:**

Yes, but the impact statement is missing.

**Strengths And Weaknesses:**

Soundness: The proofs provided by the paper are all correct and support the theoretical claims of the paper. The authors are appropriately cautious when comparing their parametrization to fully connected neural networks, acknowledging that it corresponds to wide, shallow architectures and may not share the empirical advantages of deep models. However, the connection between the abstract theory and the experimental methodology is not entirely clear. In particular, the paper does not specify how the proposed parametrization is concretely implemented in the experiments.
Presentation: The paper is generally well written and mathematically clear. The theoretical development is easy to follow, but the overall narrative could be more cohesive. The comparison to fully connected layers feels somewhat disconnected from the rest of the paper, and its relevance to the application section is not clear. More explicit discussion of how the theoretical framework translates into the experimental setup would improve clarity.
Significance: The contribution is primarily theoretical and targets a specialized but relevant problem in function approximation. While the immediate practical impact on general machine learning practice is limited, the results may be useful for future work on learning under convexity or generalized convexity constraints.
Originality: The paper offers a novel treatment of finitely Y-convex functions as a parametrization tool, along with new uniform approximation results. The links to neural networks and applications are conceptually interesting but left at a conceptual level.

---

> ### Author Rebuttal · Authors · 2026-03-25
>
> Thank you for the careful reading and for the helpful feedback. Below I address the main concerns.
>
> ## 1. How the parametrization is used in the experiments
>
> I agree that this is a valid concern; The paper should be more explicit that the finitely $\Phi$-convex parametrization is exactly the model used in the experiments.
>
> In optimal transport, I learn the Kantorovich dual potential $f \in \mathcal C^Y(X)$ using
> $$
> f(x)=\max_{i=1,\dots,n}\{\Phi(x,y^i)-p_i\},
> $$
> where the trainable parameters are the support points $y^i \in Y$ and intercepts $p_i$. In mechanism design, I learn the indirect utility $v \in \mathcal C^X(Y)$ using
> $$
> v(y)=\max_{i=1,\dots,n}\{\Phi(x^i,y)-p_i\},
> $$
> where the trainable parameters are the support points $x^i \in X$ and intercepts $p_i$. For training, I replace the hard max by a smoothed max. I then optimize the objectives given in Section 4 by replacing the expectations with finite-sample averages and applying first-order methods. In optimal transport, the transport map is then recovered from the optimized Kantorovich potential; in mechanism design, the allocation rule is recovered from the optimized indirect utility, via the formulas I already give in the paper. I will add a short implementation section to make this explicit.
>
> ## 2. Status of the equi-semiconvexity assumption in the experiments
>
> In the experiments, this assumption is satisfied in all cases. In the optimal transport experiments, the quadratic and negative quadratic kernels are smooth on compact domains, so the sufficient condition from Remark~4.8 applies. In auction Setting A, the relevant sections are affine, hence convex and therefore $0$-semiconvex. In auction Setting B, the relevant sections are convex, and therefore again $0$-semiconvex.
>
> Meaningful practical violations arise when the relevant kernel sections fail to admit a uniform lower curvature bound. In most applications, compactness is natural since the relevant spaces are bounded. Under compactness, $C^2$ smoothness is already enough by Remark~4.8, so the practically relevant failures mainly come from non-smooth kernels with downward kinks.
>
> Moreover, when $\Phi$ is genuinely non-smooth, gradient approximation is often not the primary target in the applications considered here, since the recovery formulas for the transport map and allocation rule already rely on differentiability of $\Phi$.
>
> As for the behavior when the assumption fails, the counterexample I give in the paper shows that it can fail badly: uniform approximation of the potential need not imply approximation of its gradient. I do not provide a quantitative bound for that regime, since its severity depends on the specific practical setting, in particular on how strongly the kernel sections violate a uniform lower curvature bound and on which branches are active in representing the relevant $Y$-convex function.
>
> ## 3. Cohesion and the neural-network comparison
>
> I agree that this comparison is secondary and currently too detached from the main narrative. I include it mainly for pedagogical reasons: many readers will already be familiar with maxout-type architectures, so this analogy gives a quick way to understand and remember the finitely $\Phi$-convex parametrization used in the paper. Concretely, the model computes a max (or smoothed max) over branch functions of the form $\Phi(x,y^i)-p_i$. In the inner-product case, this reduces to a max-affine / maxout-type shallow architecture; for general $\Phi$, the branches are application-specific and need not be affine. I will shorten this discussion and make clear that it is a structural analogy meant to clarify the parametrization used in the experiments, not a central claim of the paper.
>
> ## 4. Impact statement
>
> I agree that the current submission does not contain a separate impact statement. I will add the following to the camera-ready version:
>
> This paper is primarily theoretical and studies generalized convex parameterizations together with controlled simulated applications to optimal transport and mechanism design. In particular, the mechanism design settings considered here are stylized mathematical abstractions of economic interactions. Any mechanism that is "optimal" in this framework is optimal only relative to the specified primitives and objective, such as the surplus kernel, type distribution, production cost, and seller revenue criterion; this does not imply fairness, transparency, or broader social desirability. The paper does not study deployment questions or ethical constraints, and the experiments are limited to controlled synthetic settings rather than real-world decision systems.

---

> > ### Author Rebuttal · Reviewer_Gq1V · 2026-04-04
> >
> > My concerns have been resolved.

---

### Official Review · Reviewer_XeHW · 2026-03-13

**Soundness:** 3
**Presentation:** 3
**Significance:** 3
**Originality:** 3
**Overall Recommendation:** 4
**Confidence:** 3

**Summary:**

The paper introduces a differentiable parameterization for generalized convex functions (GCFs) with a convex parameter space and proves universal approximation results for both GCFs and their gradients. The main theoretical results show that finitely $Y$-convex functions are dense in the space of $Y$-convex functions and, under an equi-semiconvexity assumption on the kernel sections $x \mapsto \Phi(x,y)$, their gradients also form universal approximators for gradients of GCFs. The paper further proposes a smoothed log-sum-exp version of the construction and interprets the parameterization as a shallow neural architecture with max aggregation. On the application side, the framework is used to reformulate certain optimal transport and multi-item auction design problems as single-level optimization problems over GCFs.

**Compliance With Llm Reviewing Policy:**

Affirmed.

**Key Questions For Authors:**

1. The proposed representation uses a finite set $\tilde Y$ and approximates generalized convex functions through a maximization over elements of this set. In practice, how sensitive is the method to the size of $\tilde Y$? For example, do the experiments suggest that the approximation quality improves steadily as $|\tilde Y|$ increases, or does performance saturate quickly?

2. It is not completely clear from the experimental section whether the numerical experiments use the smoothed version or the original max formulation for the max operator. Could the authors clarify which formulation is used in the experiments? If the smoothed version is used, it would also be helpful to briefly describe how the smoothing parameter is chosen in practice. Does the choice between these two formulations affect the experimental results?


3. The paper focuses on optimal transport and mechanism design as motivating applications. Are there other classes of problems where the authors expect this generalized convex parameterization to be particularly useful?

**Limitations:**

Yes

**Strengths And Weaknesses:**

### Strengths

**Soundness.**
The core approximation results are mathematically clear and well scoped. Theorem 4.3 establishes density of finitely $Y$-convex functions in $C^Y(X)$, and Theorem 4.9 extends this to gradients under the additional equi-semiconvexity assumption. The semiconvexity-based argument is a reasonable way to recover gradient stability. The smoothed approximation result in Theorem 4.10 also fits naturally with the construction.

**Presentation.**
The paper is generally readable and the high-level motivation is easy to follow. The transition from classical convexity to generalized convexity is well motivated, and the applications to optimal transport and mechanism design help explain why one might want a parameterization of GCFs rather than a generic neural network.

**Significance and Originality.**
The paper proposes a differentiable parameterization of generalized convex functions that can be trained using gradient-based optimization. Such structures arise naturally in certain structured optimization problems, including optimal transport with non-standard kernels and mechanism design settings. The framework suggests that in these settings some problems traditionally formulated as bilevel or min–max problems may be rewritten as single-level optimization problems over generalized convex functions. The main conceptual contribution is extending universal approximation ideas from classical convex functions to generalized convex functions, including approximation of their gradients under additional regularity assumptions on the kernel. The proposed representation also has an interpretation as a shallow max-aggregation architecture, which provides some intuition for its computational implementation.

### Weaknesses

**Soundness.**
The theoretical results are correct as stated, but their scope depends on assumptions whose practical reach could be discussed more explicitly. In particular, the gradient approximation theorem requires the family of sections $x \mapsto \Phi(x,y)$ to be equi-semiconvex. Remark 4.8 gives a sufficient condition via $C^2$ regularity and compactness, but in applications the paper does not clearly identify when this assumption is easy to verify for a given kernel beyond the smooth examples considered.

A second point concerns the approximation result itself. The density theorem shows that finitely $Y$-convex functions can approximate generalized convex functions arbitrarily well, but the paper does not provide guidance on how large the finite set $\tilde Y$ may need to be in practice to obtain a good approximation. While such quantitative estimates are often difficult and are also largely unknown even for standard neural network approximation results, some empirical illustration of how approximation quality improves as $|\tilde Y|$ increases would be helpful.

A third soundness-related point is empirical rather than theoretical. The experiments illustrate that the learned maps and mechanisms behave plausibly, but they do not directly test the approximation theorems themselves. For example, one could compare approximation error as the size of the finite set $\tilde Y$ grows, or examine whether the smoothed log-sum-exp model empirically approaches the hard-max model in a controlled setting. At present, the experiments mainly demonstrate usefulness on downstream tasks rather than validate the approximation claims quantitatively.

**Presentation.**
Some mathematically important assumptions are introduced rather quickly. In the applications, the twist condition is crucial for converting gradients into transport maps or allocations, yet the practical implications of this condition are not discussed in much detail.

**Significance.**
The framework is promising, but the experimental evidence is still somewhat narrow. In optimal transport, the experiments are designed on controlled synthetic distributions with user-specified kernels, and in mechanism design the benchmarks are simulated rather than drawn from standard public datasets. The authors explain why this is reasonable, and that explanation is fair, but it still leaves open how robust the method is outside these controlled settings.

**Originality.**
The contribution appears to lie mainly in extending ideas from convex-function approximation to the setting of generalized convexity and establishing the corresponding approximation results. The paper does not propose a fundamentally new architecture or optimization method, but instead studies how generalized convex functions and their gradients can be represented and approximated within this framework.

---

> ### Author Rebuttal · Authors · 2026-03-26
>
> Thank you for the careful reading and for the constructive feedback. I address the main concerns and questions below.
>
> ## 1. Size of $\tilde Y$ and approximation errors
> I answer this point in three parts: value approximation, gradient approximation, and empirical validation.
>
> ### Value approximation
> What matters is the covering number of $Y$, namely $N_Y(r)$, the minimum number of balls of radius $r$ needed to cover $Y$.
>
> In the proof of Proposition~4.1, I use the fact that a $Y$-convex function inherits the Lipschitz constant $\lambda$ of the kernel $\Phi$. The proof takes $\tilde{Y}$ to be the centers of balls of radius $\varepsilon/(2\lambda)$ covering $Y$, so
> $$
> |\tilde Y| \le N_Y\left(\frac{\varepsilon}{2\lambda}\right).
> $$
> Since $Y$ is a compact subset of Euclidean space, this gives the crude Euclidean scaling
> $$
> |\tilde Y| = O\left(\left(\frac{\lambda}{\varepsilon}\right)^{\dim Y}\right)
> $$
> up to the scale of $Y$. If the relevant maximizing points lie on a lower-dimensional manifold, or more generally in a subset of smaller metric entropy, then the exponent is the intrinsic dimension rather than the ambient one, which may be much smaller in practice.
>
> ### Gradient approximation
> I do not currently claim a uniform $f$-independent rate for gradient approximation. The difficulty is that $\nabla f(x)$ depends on the active maximizer $y^*(x)$, not just on the value of the supremum. Uniform approximation in value does not control how stable that maximizer is, and for general $Y$-convex functions the maximization problem can be arbitrarily flat near the optimum. This is why semiconvexity gives convergence of gradients but does not, by itself, yield an explicit uniform rate.
>
> ### Empirical validation
> In the current experiments, I choose the number of support points linearly in dimension across tasks. I agree that the present experiments emphasize downstream task performance rather than directly measuring approximation error as $|\tilde Y|$ grows. Because the target optimum is not known a priori, I can only compare the solutions I recover as $|\tilde Y|$ increases.
>
> I will add a short discusison of these to the paper.
>
> ## 2. Verifying the Equi-semiconvexity Assumption
> In the experiments, I can verify this assumption directly. In the optimal transport experiments, the quadratic and negative quadratic kernels are smooth on compact domains, so the sufficient condition from Remark~4.8 applies. In mechanism-design Setting A, the relevant sections are affine, hence convex and therefore $0$-semiconvex. In mechanism-design Setting B, the relevant sections are convex, and therefore again $0$-semiconvex. I will update the paper to explain this.
>
> For an arbitrary nonsmooth kernel, I agree that verification can be difficult. That said, in the application families I study, the kernels are almost always smooth as that's how existence of transport maps and allocation maps are gauranteed. So this is not a major problem.
>
> ## 3. Ambiguity of max/softmax in Application
> You are right that the current draft does not state this clearly enough. In training, I tried three strategies: hard max, soft max, and a straight-through estimator. The results reported in the paper use a straight-through estimator during training. What worked best empirically was to start with a low softmax temperature and increase it over the course of training.
>
> At evaluation time, I use hard max. I also checked replacing hard max at evaluation with soft max, and the results were nearly identical, because by the end of training the temperature is high enough that the soft max is effectively the same as hard max. I will add this implementation detail to the revision.
>
> ## 4. Scope of Potential Applications
> Beyond optimal transport and mechanism design, closely related areas such as matching, screening, and many quasilinear pricing problems also fit naturally into this framework, since they can be phrased as optimal transport and mechnism design. In the paper, I focused on optimal transport and mechanism design because these are the two applications I know best and could therefore develop most concretely.
>
> More generally, the framework applies whenever the task can be reduced to learning an $\Phi$-convex function
> $$
> v(y)=\sup_{x\in X}\{\Phi(x,y)-g(x)\},
> $$
> Another example is inverse problems with a known forward operator, such as deblurring, tomography, or compressed sensing. Then learning the reconstruction prior or regularizer $g(x)$ is equivalently learning the induced generalized convex value function $v$.
>
> ## 5. Twist Condition in the Applications
> I agree that twist condition is glossed over quickly. Now that I have one more page, I will explain the practical role of the twist condition more explicitly. In the applications, twist is the injectivity condition that lets me recover the underlying decision rule from the gradient of the learned potential: without it, the potential can still be learned, it may not induce a transport map or allocation rule.

---

> > ### Author Rebuttal · Reviewer_XeHW · 2026-04-03
> >
> > Given that the authors have committed to including these clarifications and discussions in the final version, I am satisfied with the rebuttal and maintain the current score.

---

### Decision · Program_Chairs · 2026-04-30

**Decision:**

Accept (regular)

**Comment:**

This paper introduces a differentiable parameterization of generalized convex functions (GCFs) with a convex parameter space, and establishes universal approximation results both for GCFs and, under an equi-semiconvexity assumption, for their gradients. The paper further demonstrates how this representation can be used in applications such as optimal transport and multi-item mechanism design to reformulate certain bilevel or min-max problems as single-level optimization problems amenable to first-order methods.

The reviewers were generally positive about the paper’s core theoretical contribution. The main concerns raised during review were about clarity and scope rather than correctness. For instance: i) how the finitely (Y)-convex parameterization is actually used in the experiments; ii) more explicit discussion of the equi-semiconvexity and assumptions; iii) a tighter connection between the abstract theory and the experimental implementation.

Overall, the paper makes a meaningful theory contribution with credible supporting applications. While the empirical evaluation is not broad and the paper would benefit from a clearer presentation of its implementation pipeline and limitations, these issues appear addressable in the final version and do not outweigh the value of the main theoretical results.

For the final version, we encourage the authors to make the experimental pipeline fully explicit, clarify the practical scope of the assumptions, and streamline the discussion of the neural-network connection and broader limitations.